# SAM/SAH Mediates Parental Folate Deficiency-Induced Neural Cell Apoptosis in Neonatal Rat Offspring: The Expression of Bcl-2, Bax, and Caspase-3

**DOI:** 10.3390/ijms241914508

**Published:** 2023-09-25

**Authors:** Qinghan Ren, Guoquan Zhang, Ruiting Yan, Dezheng Zhou, Li Huang, Qianwen Zhang, Wen Li, Guowei Huang, Zhenshu Li, Jing Yan

**Affiliations:** 1Department of Nutrition and Food Science, School of Public Health, Tianjin Medical University, Tianjin 300070, China; renqinghan0223@tmu.edu.cn (Q.R.); zgq1217@tmu.edu.cn (G.Z.); yanruiting1222@tmu.edu.cn (R.Y.); dezhengzhou@163.com (D.Z.); huangli123@tmu.edu.cn (L.H.); z18522879458@163.com (Q.Z.); liwen828@tmu.edu.cn (W.L.); huangguowei@tmu.edu.cn (G.H.); 2Tianjin Key Laboratory of Environment, Nutrition and Public Health, Tianjin 300070, China; 3Department of Social Medicine and Health Administration, School of Public Health, Tianjin Medical University, Tianjin 300070, China

**Keywords:** folate, parental, apoptosis, Bcl-2, Bax, Caspase-3, neural cells, SAM, SAH

## Abstract

Research demonstrated that folate deficiency in either the mother or father could impact the biological functions of the offspring’s of neural cells. Folate deficiency can also impair the methionine cycle, thus contributing to the conversion of S-adenosylmethionine (SAM) to S-adenosylhomocysteine (SAH), which could potentially cause damage to the central nervous system. The study focused on the effect of parental folate deficiency on neural cell apoptosis in offspring neonatal rats and whether it is mediated by the levels of SAM and SAH in brains. The experimental design was conducted by feeding female and male Sprague Dawley (SD) rats with either folate-deficient or folate-normal diets, sacrificing the offspring within 24 h and isolating their brain tissue. Rats were divided into four groups: the maternal-folate-deficient and paternal-folate-deficient (D-D) group; the maternal-folate-deficient and paternal-folate-normal (D-N) group; the maternal-folate-normal and paternal-folate-deficient (N-D) group; and the maternal-folate-normal and paternal-folate-normal (N-N) group. There was down-regulation of B-cell lymphoma 2 (Bcl-2) expression, up-regulation of Bcl-2-associated X protein (Bax) and Caspase-3 expression of neural cells, and pathological changes in the brain ultrastructure, as well as decreased SAM levels, increased SAH levels, and a decreased SAM/SAH ratio in the rat fetal brain via parental folate deficiency. In conclusion, parental folate deficiency could induce the apoptosis of neural cells in neonatal offspring rats, while biparental folate deficiency had the greatest effect on offspring, and the unilateral effect was greater in mothers than in fathers. This process may be mediated by the levels of SAM and SAH in the rat fetal brain.

## 1. Introduction

Folate, also known as vitamin B9, is a water-soluble vitamin that is essential for biological processes in the body, and it is implicated in DNA synthesis and repair, the methionine cycle, and methylation processes [1]. Maternal folate status has been widely demonstrated to perform a vital function in pregnancy and fetal brain development due to its requirement for neural cell differentiation and division and tissue growth [2]. Increasing evidence also indicated that such an association existed between paternal folate intake prior to conception and offspring brain health and disease. A study in rats emphasized that adult offspring with paternal folate deficiency were more susceptible to developing anxiety and depressive traits [3]. Paternal red blood cell folate levels during the periconceptional period have been suggested to be involved in embryonic development by a prospective cohort study [4]. Although either paternal or maternal folate levels during pregnancy and preconception have been repeatedly described as exerting an effect on offspring development, the combined effects on the fetal brain and the mechanisms involved are not well established.

Embryonic development is a pivotal stage in generating and organizing neurons in the brain [5]. Apoptosis, a programmed form of cell death, serves an important function in eradicating aberrant cells that threaten development, homeostasis, and overall survival; whereas abnormal apoptosis of neurons during embryonic development may have implications for the normal morphogenesis and function of the nervous system [6]. Apoptosis can be initiated via intrinsic and extrinsic pathways [7]: in the intrinsic pathway, members of the B-cell lymphoma 2 (Bcl-2)family proteins play a pivotal role in regulating apoptosis by controlling pro-apoptotic and anti-apoptotic intracellular signals [8]; in the extrinsic pathway, caspases, a family of cysteine proteases, are activated by a set of proapoptotic signals. Caspase-3 is critical for executing apoptosis in the progress [9]. Increased caspase-3 activity is accompanied by changes in the Bcl-2 family, with up-regulation of the pro-apoptotic protein Bcl-2-associated X protein (Bax) and down-regulation of the anti-apoptotic protein Bcl-2 [10]. Experimental studies in mice by Shao Y et al. revealed that the disruption of folate metabolism in the central nervous system could induce astrocyte apoptosis [11]. Folate deficiency in mothers in pregnancy may cause hippocampal dysfunction in offspring through neuronal apoptosis [12]. While paternal folate deficiency has been demonstrated to affect offspring neurodevelopment [13], the relationship between paternal folate levels and neuronal apoptosis in the fetal brain is rarely reported. This study investigated the role of maternal and paternal folate deficiency on the neural cells’ apoptosis in offspring neonatal rats by assaying the expression of Bcl-2 family proteins and Caspase-3.

S-adenosylmethionine (SAM) is a crucial metabolic intermediate and a versatile methyl donor required for the cellular synthesis and modification of many biomolecules, including nucleic acids, proteins, and hormones [14]. SAM is produced via the methionine cycle and its demethylation leads to S-adenosylhomocysteine (SAH), a precursor of homocysteine (Hcy) [15]. Folate deficiency impels SAM to convert to SAH, which is then converted to Hcy and decreases the SAM/SAH ratio and DNA methylation potential [16]. Findings from mice research suggest that maternal vitamin D deficiency reduced serum folate concentrations and cortical SAM levels in offspring, while folic acid supplementation during pregnancy alleviated offspring’s cortical SAM depletion [17]. Yang et al.’s population-based study also suggested that low folate concentration in mothers was associated with decreased SAM concentrations and increased SAH levels in their offspring with neural tube defects [18]. However, the current studies present no direct evidence indicating that such an association exists between fathers or both parents and the offspring.

Furthermore, SAM is involved in additional metabolic pathways that require methyl to regulate cell growth and survival [19]. Experimental results in infected Wistar rats indicated that dysregulation of methionine homeostasis led to Hcy accumulation, SAM level reduction (SAM/SAH ratio reduction), and consequently apoptosis of the hippocampal dentate gyrus of infant rats [20]. Findings in the Drosophila model also suggested that the depletion of SAM in chromosomal instability cells led to cellular apoptosis, accompanied by gene changes affecting one-carbon metabolism [21]. As mentioned, folate levels can modulate SAM concentrations, consequently altering neural cell function and quantity [22]. Cognitive decline and increased aggression were found in mice with folate deficiency and/or genetically impaired folate utilization, which may be related to reduced SAM via folate deficiency [23]. Nevertheless, whether dietary folate levels in the parents affect neuronal apoptosis by modulating SAM and SAH levels in the neonatal offspring rat brain it is not well understood.

Based on the previous evidence, the present study hypothesized that parental folate deficiency could induce neuronal apoptosis in newborn rat offspring, accompanied by regulation of the expression of Bcl-2 family proteins and Caspase-3, in which modulation of the SAM and SAH levels may operate as mediators of the process.

## 2. Results

Our previous study demonstrated reduced folate levels in the neonatal rat offspring’s brain tissue, while parental folate deficiency increased Hcy concentrations. Furthermore, the N-N group demonstrated higher folate levels and lower Hcy concentrations in offspring brain tissue than the D-N group [24].

### 2.1. Parental Folate Deficiency Decreased the Expression of Bcl-2 and Increased the Expression of Bax of Neural Cells in the Brain of Neonatal Offspring Rats

To determine whether parental folate deficiency led to changes in neuronal apoptosis in the fetal brain through intrinsic pathways, we examined the expression of Bcl-2 family proteins (Bcl-2, Bax) via immunofluorescence. The expression of Bcl-2 in the hippocampus manifested lower in the D-D group than in the N-N group (Figure 1b, *p* < 0.05). In the cerebral cortex, Bcl-2 expression was up-regulated in the N-D and N-N groups compared to the D-D group and down-regulated in the D-N group compared to the N-N group (Figure 1c, *p* < 0.05). In the fetal rat hippocampus, a greater number of Bax-positive cells were observed in the D-D group than in the N-D and N-N groups, with a surplus in the D-N group over the N-N group, while the N-N group had the fewest Bax-positive cells in the cerebral cortex among the four groups. (Figure 2, *p* < 0.05).

The experimental findings suggested that reduced anti-apoptotic protein Bcl-2 expression and improved pro-apoptotic protein Bax expression via by parental folate deficiency were observed in pups’ hippocampus and cerebral cortex neural cells. The impacts of both maternal and paternal folate deficiency on the expression of Bcl-2 and Bax in offspring’s neurons were greatest together and greater unilaterally in the maternal generation than in the paternal generation.

### 2.2. Parental Folate Deficiency Increased Expression of Caspase-3 of Neural Cells in the Brain of Neonatal Offspring Rats

To further investigate whether folate insufficiency in the parents could affect fetal neuronal apoptosis via the extrinsic pathway, the expression of the pro-apoptotic protein Caspase-3 was determined in the neonatal rat hippocampus and cerebral cortex via immunofluorescence. Figure 3 showed that Caspase-3 in neural cells’ expression in the offspring rat hippocampus was greater in the D-D group than in the N-N group. Meanwhile, in the cerebral cortex, higher Caspase-3 expression was apparent in the D-D group compared to the N-D and N-N groups, with a lower level in the N-N group than in the D-N group (*p* < 0.05). Our observations revealed that parental folate deficiency stimulated Caspase-3’s expression in the neural cells of fetal rat brains. In addition, both parental folate deficiencies exerted effects on the offspring, with biparental folate deficiency having the greatest effect and unilateral folate deficiency exhibiting a stronger effect in females than in males.

### 2.3. Parental Folate Deficiency Induced Pathological Changes in the Ultrastructure in Neuronal Cell Bodies of Neonatal Offspring Brain

As shown in Figure 4, the N-N group appeared to have normal mitochondrial structures, with a typical homogeneous staining pattern of the matrix, a clear visible double membrane and tight orderly cristae, and myelinated axons with a compact layer of myelin lamellae and a greater number of synapses (*p* < 0.05). In contrast, the obvious pathological changes in neuronal cell bodies in the D-D and D-N groups showed swollen mitochondria, exhibiting expanded matrix space, fragmented or disorganized cristae, damaged membrane structures, decompaction myelin exhibiting myelin sheaths surrounding axons loosening with widespread, slightly disrupted layers of myelin lamellae and vacuoles, as well as a smaller number of synapses (*p* < 0.05). Evidence from transmission electron microscopy (TEM) experiments suggested that parental folate deficiency induced pathological changes in the ultrastructure of the hippocampus in neonatal offspring brains.

### 2.4. SAM and SAH Levels in the Brain Tissue of Offspring Neonatal Rats

The SAM and SAH concentration in fetal brain tissue was detected via the high-performance liquid chromatography (HPLC) technique. Compared with the N-N group, parental folate deficiency reduced SAM levels and increased SAH levels at postnatal day 0 (PND0) in offspring brain tissue. A higher SAM concentration and lower SAH concentration were shown in the N-D group than in the D-D group (*p* < 0.05, Figure 5a,b). Moreover, the SAM/SAH ratio was lower in the D-D group than in the N-D and N-N groups, while it was higher in the N-N group than in the D-N group (*p* < 0.05, Figure 5c). The results demonstrated that parental folate deficiency reduced SAM concentration and improved the SAH concentration of offspring rat brain tissue at PND0, thus reducing the SAM/SAH ratio. Furthermore, parental folate deficiency had the strongest effect on the offspring, while maternal folate deficiency had a more adverse effect when only one parent was folate deficient.

## 3. Discussion

The findings of the present study showed that parental folate deficiency induced neuronal apoptosis in offspring neonatal rats, demonstrated by the down-regulated expression of the anti-apoptotic protein Bcl-2, the up-regulated expression of the anti-apoptotic proteins Bax and Caspase-3 of neural cells in the hippocampus and cerebral cortex, and pathological changes in the ultrastructure of neuronal cell bodies in offspring. This effect of parental folate deficiency was associated with decreased SAM and increased SAH concentrations (reduced SAM/SAH ratio) in the offspring’s brain tissue.

The proper balance between neural cell proliferation, apoptosis, and cell cycle dysregulation is crucial for neurodevelopment, neurodegeneration, and neurological disorders [25]. Maternal dietary folate regulates the fetal brain’s neurogenesis and neuronal apoptosis [26]. Craciunescu CN et al. found that folic acid deficiency during late pregnancy reduced the proliferation and improved the progenitor cells’ apoptosis in the fetal brain of mice [27]. Consistent with the previous findings, our present data also indicated increased neuronal apoptosis in offspring neonatal rats due to parental folate deficiency. However, we further explored alternative potential mechanisms by which feeding parents folate-deficient diets increased neural cell apoptosis in pups’ brains via Bcl-2/Bax/Caspase-3 pathways.

Bcl-2, Bax, and Caspase-3 are pivotal modulators of the apoptotic pathway [28]. Animal studies showed that maternal folate deficiency down-regulated the expression of the anti-apoptotic proteinBcl-2 and up-regulated the expression of the pro-apoptotic proteinBax, while also increasing the ratio of cleaved Caspase-3/Caspase-3 and Caspase-3 activity in fetal brains’ neural cells [29]. In line with the above-mentioned results, apoptosis induced via parental folate deficiency may be confirmed by the down-regulation of Bcl-2 expression, the up-regulation of Bax expression, and the activation of a caspase-dependent signaling pathway in the present study. Moreover, folate deficiency during pregnancy has been identified in previous animal studies to cause pathological changes in the hippocampus and cerebral cortex of the offspring [30,31]. Our experimental findings similarly illustrated that parental folate deficiency contributed to ultrastructural damage in the hippocampus of offspring neonatal rats. Taken together, this present study provided compelling evidence that parental folate deficiency could improve neuronal apoptosis of the fetal rat brain.

Furthermore, in this cross-generation rat experiment, we examined SAM and SAH concentrations in offspring neonatal rats’ brain tissue via HPLC and obtained evidence that parental folate deficiency reduced SAM concentration and increased SAH concentration in the brain tissue of offspring neonatal rats (thus reducing SAM/SAH ratio). A possible explanation for the results may be that folate deficiency results in impaired methionine synthesis, which leads to depletion of SAM and elevation of SAH (i.e., increased SAM/SAH ratio), thus contributing to Hcy synthesis [32]. This linkage may be established during embryonic development and is mediated by maternal folate intake through the one-carbon metabolism [33], which is considered one of several epigenetic pathways that regulate genomic programming and brain development [34]. Maternal folate status has been reported to influence folate-mediated one-carbon metabolism, accompanied by a decrease in SAM levels and an increase in SAH levels in the placenta of rats [35]. It was proposed in a Brazilian study that a low ratio of SAM to SAH is related to folate deficiency in pregnant women and newborns [36]. Nevertheless, there is currently little to no evidence of this epigenetic link between fathers and their offspring, whereas this rat study confirmed the existence of such as link between parents and neonates.

Research we have previously conducted in rats demonstrated that parental folate deficiency decreased folate levels and increased the Hcy levels of the offspring’s brain tissue [24]. Increased Hcy in fetal rat brain tissue may lead to decreased recirculation of Hcy to SAM and increased SAH levels, resulting in a decreased methylation capacity, potentially altering the biological function of the central nervous system [37]. The literature supports the theory that the disruption of methionine/SAM metabolism in the established mixed lineage leukemia (MLL)-AF4 cell line reduced the overall cellular methylation potential, diminished relative cell numbers, and selectively induced apoptosis [38]. Moreover, experimental studies indicated that folic acid could protect hepatocytes from apoptosis in hyperhomocysteinemia mice through reduced SAM and SAH cellular concentrations and increased SAM/SAH ratio-mediated methylation [39]. Consistent with these findings, parental dietary folate insufficiency may stimulate neuronal apoptosis in offspring by adjusting the offspring brain tissue levels of circulating methionine metabolites such as Hcy, SAM, and SAH.

Most study models have been single-parent designs, and the differences between the combined effects of both parental folate deficiencies and the effect of unilateral folate deficiencies are inconclusive. Our observations in this study demonstrated that folate deficiency in both the mother and father affected neuronal apoptosis mediated by altered SAM and SAH concentration in rat fetal brains, with the greatest effect being produced by both sexes acting together. The effect of single-sex folate deficiency in both the paternal and maternal progeny was stronger in the females than in the males. The reason for this may be the difference between paternal and maternal epigenetic inheritance, as maternal epigenetic reprogramming may occur at different times, more directly altering the offspring’s environment (periconceptional environment) and affecting early embryonic reprogramming, whereas paternal epigenetic inheritance can only be transmitted via sperm [40].

Since the 1980s, attention has been focused on the effects of maternal folate deficiency during pregnancy on the fetal nervous system [41]. Adequate folic acid intake is particularly critical for pregnant women, whose folic acid requirements are 5–10 times higher than those of non-pregnant women. These increased requirements are essential to supporting the optimal growth and development of maternal and fetal tissues [42]. Maternal folate deficiency is understood to impact the neurodevelopment and brain health of the offspring, potentially leading to neurodevelopmental disorders, including neural tube defects [43]. However, the mechanisms by which maternal folate levels affect offspring’s neurodevelopment, including DNA methylation [44], oxidative stress [45], inflammation [46], etc., are complex and inconclusive. Nevertheless, the relationship between paternal folate intake and fetal brain development has very rarely been reported. The findings of animal experiments demonstrated that paternal dietary folate deficiency may adversely affect birth outcomes in mice via epigenetic transmission involving sperm histone H3 methylation or DNA methylation, while dietary folate supplementation is critical for the offspring’s health [47]. Aatish Mahajan et al. demonstrated that an unbalanced folate in the paternal generation’s diet could disrupt the progeny’s methionine cycle [48]. Our findings also revealed that folate deficiency in either the paternal or maternal generation has consequences for the fetal nervous system and that SAM/SAH may be a regulatory factor. There are certainly other unexplored pathways, so more attention should be paid in the future to the association between both maternal and paternal folate intake and offspring health and the mechanisms involved.

The study was a reproduction experiment in SD rats, with interventions in parental female and male rats fed folate-deficient and folate-normal diets, and related experiments were conducted to investigate the effects of folate-deficient intake by a single parent and both parents on the offspring’s nerve cell apoptosis. Limitations of this research also exist. There must be multiple molecular mechanisms linking maternal and paternal folate levels to the offspring’s nervous system, and more detailed and comprehensive studies should be designed to explore the contributing factors and, in doing so, establish the basis for folate intake by both parents in the periconceptional period, as well as for the prevention of neurological disorders in newborns.

## 4. Materials and Methods

### 4.1. Animals and Dietary Treatment

The study is a reproduction experiment in SD rats, which are commonly used experimental animals for their adaptability to the environment, rapid growth and development, high reproduction rate, and consistent genetics, making them the preferred rodent for reproduction study [49]. Forty-eight 6-week-old SD female rats and twenty-four 6-week-old SD male rats (Charles River Laboratories, Beijing, China) were assigned into four groups according to their feeding patterns: (1) D-D group: both female and male rats were fed the folate-deficient diet; (2) D-N group: female rats were fed the folate-deficient diet, and male rats were fed the folate-normal diet; (3) N-D group: female rats were fed the folate-normal diet, and male rats were fed the folate-deficient diet; (4) N-N group: both female and male rats were fed the folate-normal diet. Females and males were mated for two weeks, during which time females were fed the same diet as previously, and males in the same cage were fed the same diet as the females. Rats were housed in a specific pathogen-free facility under a controlled 12-h light/dark cycle and provided food and water ad libitum. All animal procedures were approved by the Tianjin Medical University Animal Ethics Committee (TMUaMEC2021027).

The folate-deficient diet (<0.1 mg/kg, equivalent < 20 μg/d for humans) and the folate-normal diet (2 mg/kg, equivalent 400 μg/d for humans) modified from the AIN-93G diet were both obtained from the Trophic Animal Feed High-Technology Company (Nantong, Jiangsu, China). After delivery, the offspring rats were sacrificed at PND0, and the brain tissue was removed. Some of the brain tissues (with cerebellum removed) were either fixed with 4% paraformaldehyde (PFA), processed into paraffin sections, or collected quickly with liquid nitrogen snap freezing and stored at −80 °C for future assays. The other brain tissue (right hippocampus) was fixed with 2.5% glutaraldehyde for TEM analysis, and the corresponding left brain tissue was flash-frozen and stored as mentioned.

### 4.2. Immunofluorescence Analysis

An immunofluorescence assay was conducted as described previously [24]. The offspring brain tissue was immediately immersed in 4% PFA for 48 h, then dehydrated and embedded in paraffin. The paraffin blocks were later cut into 3 μm thin slices for attachment to the glass slide. Then, the slices were soaked sequentially in xylene for 15 min twice, and with 100% ethanol, 90% ethanol, 75% ethanol, and 50% ethanol for 10 min each. After inhibiting endogenous peroxidase activity with 3% H_2_O_2_ for 10 min and permeabilizing with 0.5% Triton X-100 for 15 min, the slices were microwave-repaired in sodium citrate buffer until the liquid boiled. Then, they were physically cooled down to room temperature and washed with phosphate-buffered saline (PBS). Non-specific binding sites were blocked with 10% goat serum for 1 h at 37 °C. Slices were then incubated with primary antibodies (rabbit anti-Bcl-2 (1:100, Abcam, Cambridge, UK); rabbit anti-Bax (1:100, Abcam, UK); and rabbit anti-Caspase-3 (1:100, Abcam)) overnight at 4 °C. The next day, the slices were removed, and the remaining primary antibody was washed with PBS, followed by incubation with goat anti-mouse secondary antibody (tetramethyl rhodamine isothiocyanate (TRITC)-conjugated anti-rabbit antibody, 1:100, Sparkjade, Jinan, Shandong, China) at room temperature (RT) for 1 h. Finally, sections were counterstained with 4′,6-diamidino-2-phenylindole (DAPI)-containing Vectashield (Sparkjade). Immunofluorescence signals were captured using the Olympus IX81 microscope (Olympus, Tokyo, Japan) and analyzed ia Image J software (version 1.54, National Institutes of Mental Health, Bethesda, MD, USA).

### 4.3. Transmission Electron Microscopy

Hippocampal tissue samples were collected from neonatal offspring rats (*n* = 2/group) for ultrastructural examination. About 1–2 mm^3^ pieces were rapidly cut from the hippocampus in the brain, fixed with 2.5% glutaraldehyde at 4 °C for 24 h, post-fixed in 1% osmium tetroxide for 2 h at RT, dehydrated in graded alcohol, permeabilized in graded acetone series at 37 °C for 8–12 h each time, and embedded in araldite and dodecenyl succinic anhydride mixture for 48 h at 60 °C. Ultrathin sections (80–100 nm) were cut with an ultramicrotome (Leica UC7, Leica Microsystems, Tokyo, Japan), stained with 2% uranyl acetate for 30 min and lead citrate for 15 min, and then observed under Transmission Electron Microscope (Tecnai G2 20 TWIN; FEI Company/Thermo Fisher Scientific, Waltham, MA, USA).

### 4.4. SAM and SAH in Brain Tissue Detected by High-Performance Liquid Chromatography (HPLC)

The content of SAM and SAH in offspring brain tissue was measured via HPLC. A total of 15 g brain tissue was homogenized in 300 μL tissue lysate (Sparkjade) and centrifuged at 15,000× rpm for 15 min at 4 °C. The supernatant (150 μL) and 30 μL of trichloroacetic acid (400 g/L) were centrifuged at 4 °C, 20,000× rpm, for 10 min. The supernatant was filtered through a 0.22 μm filter (Millipore, Burlington, MA, USA) and injected into an HPLC system (Shimadzu Co., Kyoto, Japan) containing a Hypersil GOLDTM aQ column (Thermo Fisher Scientific, Waltham, MA, USA). For the analysis of the SAH and SAM in the brain tissue, the mobile phase consisted of 50 mM sodium phosphate monobasic (pH = 4.38), 10 mM sodium heptane sulfonate, and 20% methyl alcohol (*v*/*v*). The flow rate was 1.0 mL/min, the ultraviolet (UV) detection wavelength was 258 nm, and the column temperature was 28 °C. The elution peaks were identified using SAM and SAH standards (Sigma Aldrich, St. Louis, MO, USA). The protein concentration was determined via the bicinchoninic acid (BCA) protein quantitative Kit (Sparkjade), and then the measured SAM and SAH concentrations were normalized with the measured protein concentration.

### 4.5. Statistical Analysis

Data are expressed as mean ± SD via the statistical software package SPSS 24.0 (IBM Corp, Armonk, NY, USA). One-way analysis of variance (ANOVA) was used for multiple group comparisons, followed by the Student-Newman-Keuls (SNK) test. The difference between groups was considered significant if *p* < 0.05.

## 5. Conclusions

In conclusion, the present study demonstrated that folate deficiency in parental rats induced neuronal apoptosis in the brain of neonatal offspring rats. Both maternal and paternal folate deficiency exerted effects on neonatal rat brain apoptosis, but the combined effect was the greatest, and the effect of unilateral folate deficiency alone was more pronounced in the mother than in the father. Furthermore, down-regulated SAM levels and up-regulated SAH levels in fetal rat brain tissue may be regulators of this process. This study highlights the importance of parental folate intake for the nervous system of the offspring; thus, preconception care should shift from mother-centered to parent-centered, and the mechanisms involved need to be further explored.

## Figures and Tables

**Figure 1 ijms-24-14508-f001:**
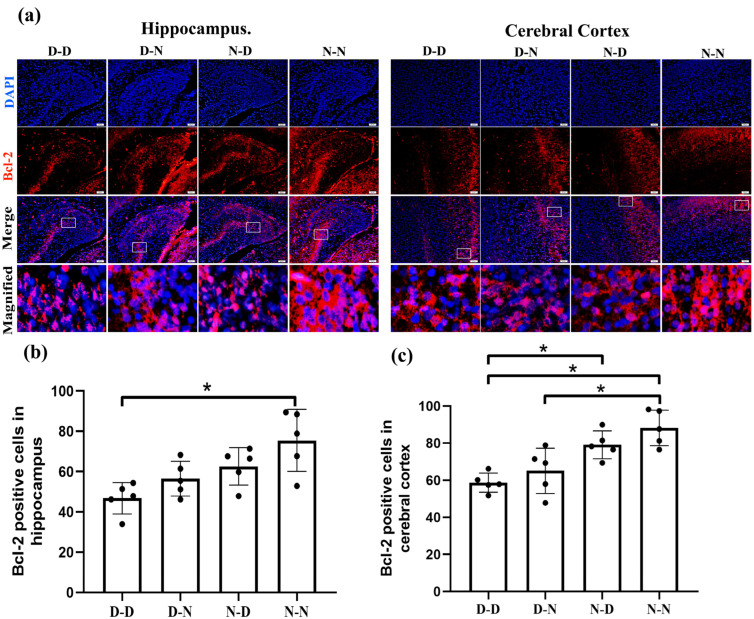
Parental folate deficiency decreased B-cell lymphoma 2 (Bcl-2) expression in neural cells in neonatal offspring rats’ hippocampus and cerebral cortex. Rats were assigned into four groups: the maternal-folate-deficient + paternal-folate-deficient (D-D) group; the maternal-folate-deficient + paternal-folate-normal (D-N) group; the maternal-folate-normal + paternal-folate-deficient (N-D) group; and the maternal-folate-normal + paternal-folate-normal (N-N) group. (**a**) Representative images of Bcl-2 expression in neural cells in the hippocampus and cerebral cortex, in which cells are stained red with Bcl-2 and blue with 4′,6-diamidino-2-phenylindole (DAPI). Scale bar = 50 μm. Each column below the merged image shows an enlarged image of the corresponding image rectangle on the merged image. (**b**) Bcl-2 expression in neural cells in the hippocampus. (**c**) Bcl-2 expression in neural cells in the cerebral cortex. Data are expressed as mean ± SD (*n* = 5). *: *p* < 0.05.

**Figure 2 ijms-24-14508-f002:**
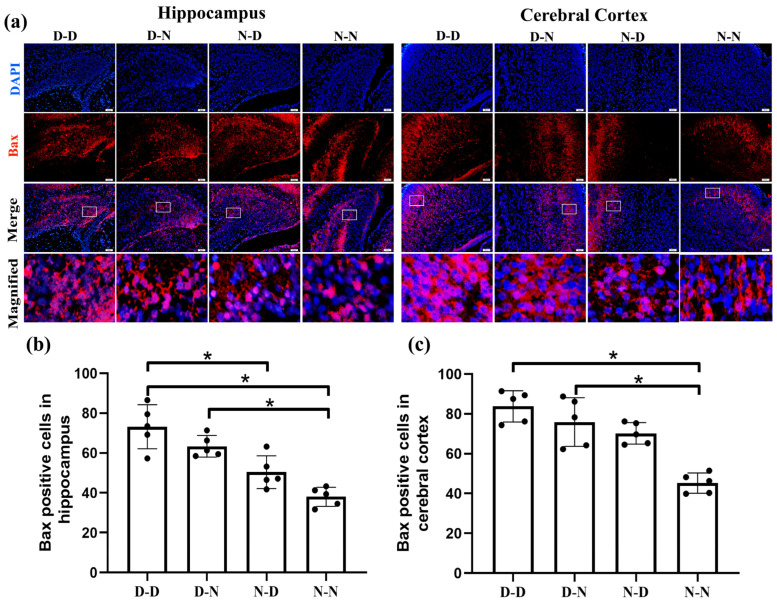
Parental folate deficiency increased Bcl-2-associated X protein (Bax) expression in neural cells in neonatal offspring rats’ hippocampus and cerebral cortex. Offspring rats were grouped as described in Figure 1. (**a**) Representative images of Bax expression in neural cells in the hippocampus and cerebral cortex, in which cells are stained red with Bax and blue with DAPI. Scale bar = 50 μm. Each column below the merged image shows an enlarged image of the corresponding image rectangle on the merged image. (**b**) Bax expression in neural cells in the hippocampus. (**c**) Bax expression in neural cells in the cerebral cortex. Data are expressed as mean ± SD (*n* = 5). *: *p* < 0.05.

**Figure 3 ijms-24-14508-f003:**
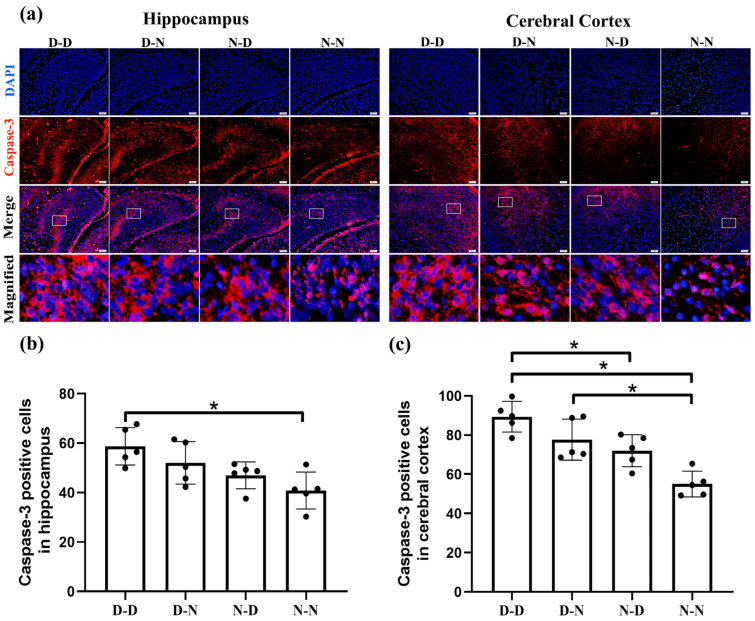
Parental folate deficiency increased Caspase-3 expression in neural cells in neonatal offspring rats’ hippocampus and cerebral cortex. Offspring rats were grouped as described in Figure 1. (**a**) Representative images of Caspase-3 expression in neural cells in the hippocampus and cerebral cortex, in which cells are stained red with Caspase-3 and blue with DAPI. Scale bar = 50 μm. Each column below the merged image shows an enlarged image of the corresponding image rectangle on the merged image. (**b**) Caspase-3 expression in neural cells in the hippocampus. (**c**) Caspase-3 expression in neural cells in the cerebral cortex. Data are expressed as mean ± SD (*n* = 5). *: *p* < 0.05.

**Figure 4 ijms-24-14508-f004:**
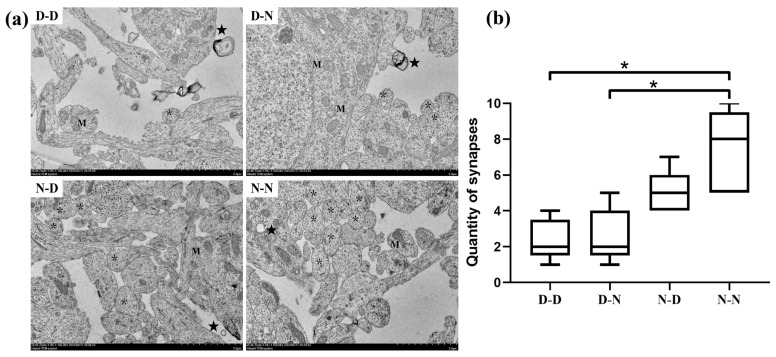
Parental folate deficiency-induced pathological changes in the ultrastructure of the hippocampal region in the offspring. Offspring rats were grouped as described in Figure 1, and hippocampal samples of offspring rats were collected for transmission electron microscope examination. (**a**) Ultrastructural changes of the hippocampal region in offspring rats. The letter M indicates swollen mitochondria exhibiting expanded matrix space, fragmented or disorganized cristae, and damaged membrane structure. The symbol ★ shows an impaired myelinated axon with decompacted myelin. The symbol * shows an intact synapse structure with a clear synaptic gap, a thickened synaptic membrane rich in dense material, and abundant synaptic vesicles in the presynaptic terminal. (**b**) Quantity of synapses. Data are expressed as median with minimum to maximum (*n* = 2, 10 visual fields/rat). Scale bar = 2.0 μm. *: *p* < 0.05.

**Figure 5 ijms-24-14508-f005:**
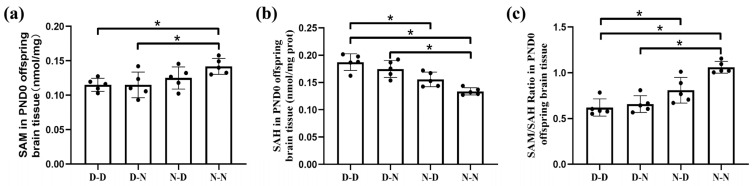
S-adenosylmethionine (SAM) and S-adenosylhomocysteine (SAH) levels in the brain tissue of offspring neonatal rats. Offspring rats were grouped as described in Figure 1. (**a**) SAM levels in the brain tissue of offspring neonatal rats. (**b**) SAH levels in the brain tissue of offspring neonatal rats. (**c**) SAM/SAH ratio in the brain tissue of offspring neonatal rats. Data are expressed as mean ± SD (*n* = 5). *: *p* < 0.05.

## Data Availability

The data presented in this.study are available on request from the corresponding author.

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
