# Peer review of "SAM/SAH Mediates Parental Folate Deficiency-Induced Neural Cell Apoptosis in Neonatal Rat Offspring: The Expression of Bcl-2, Bax, and Caspase-3"

_ijms, 2023, doi:10.3390/ijms241914508_

Round 1

Reviewer 1 Report

The study investigated the effects of parental folate deficiency on brain development. The study had 4 groups: in the first group, only the maternal diet was deficient in folate; in another group, only the paternal diet was deficient; in the third group, both parents’ diet was deficient; and in the 4th group, both patents had a normal folate level in their diet. The pups were euthanized 24 hours after birth and immune-stained, while other pups’ brains were collected for HPLC and transmission electron microscopy. They found that both parents having a deficient folate level had a negative effect on brain development, supported by both an increase in Bax-positive cells and a decrease in Bcl-2-positive cells. The maternal deficient diet group was the second worst group, followed by the paternal deficient diet group.

Recommendation:

  1. Please include high mag images for figures 1, 2, and 3. It is difficult to see the details in your magnified images. Additionally, please include higher-resolution images. When zooming in, the pictures become pixelated.
  2. In your method section, please spell out SNK in the statistical test sections.
  3. Please include more details about the housing of the animals during mating. How long were the pair together, and did they have access to food? If so, which diet was available to them?
  4. Please consider doing a TUNEL assay on your sample to demonstrate apoptosis. This will support your claims.
  5. Please consider adding an outcome where Hcy levels are investigated. It could be immunostaining, western blots, or HPLC. This will support your idea about Hcy.
  6. Please include a section about the limitations of the study in your discussion section.
  7. Additionally, please add a paragraph on how you think folate deficiency in the males would lead to changes in the offspring. Do you believe it is due to epigenetic changes?

Please review the paper again there were a few errors in the manuscript. 

one example is in the abstract with a typo in the last sentence conclusion is misspelled. 

Author Response

Point-by-point response to the reviewers’ comments:

Reviewer #1:

Comments and Suggestions for Authors

The study investigated the effects of parental folate deficiency on brain development. The study had 4 groups: in the first group, only the maternal diet was deficient in folate; in another group, only the paternal diet was deficient; in the third group, both parents’ diet was deficient; and in the 4th group, both patents had a normal folate level in their diet. The pups were euthanized 24 hours after birth and immune-stained, while other pups’ brains were collected for HPLC and transmission electron microscopy. They found that both parents having a deficient folate level had a negative effect on brain development, supported by both an increase in Bax-positive cells and a decrease in Bcl-2-positive cells. The maternal deficient diet group was the second worst group, followed by the paternal deficient diet group.

Response: Thank you for reviewing this article and providing valuable suggestions.

Comment 1-1:1.  Please include high mag images for figures 1, 2, and 3. It is difficult to see the details in your magnified images. Additionally, please include higher-resolution images. When zooming in, the pictures become pixelated.

Response 1-1: Thank you for your suggestion. As suggested, we have provided mag images with increased magnification in the Results section. Due to the limitation of the format that the total amount of data for all files must not exceed 120 MB, the images inserted in the manuscript may not be clear after compression. We also submitted the original image files when we submitted the manuscript, and we added the original images below. We hope the clarity has been improved. The figures please see the attachment. 

Comment 1-2: 2. In your method section, please spell out SNK in the statistical test sections.

Response 1-2: Thank you for the comment. We agree with the comment and have added the part as suggested. It has been indicated in red font.

Revised parts

Materials and Methods (page 11, line 385-387)

“Data are expressed as mean ± SD by the statistical software package SPSS 24.0 (IBM Corp, Armonk, NY, USA). One-way analysis of variance (ANOVA) was used for multiple group comparisons, followed by the Student-Newman-Keuls (SNK) test.”

Comment 1-3: 3. Please include more details about the housing of the animals during mating. How long were the pair together, and did they have access to food? If so, which diet was available to them?

Response 1-3: Thank you for the suggestion. Females and males were mated for two weeks, during which time females were fed the same diet as previously, and males in the same cage were fed the same diet as the females. We have added the part as suggested, and it has been indicated in red font.

Revised and added parts

Materials and Methods (page 9, line 323-326)

Females and males were mated for two weeks, during which time females were fed the same diet as previously, and males in the same cage were fed the same diet as the females. Rats were housed in a specific pathogen-free facility under a controlled 12-h light/dark cycle and provided food and water ad libitum.”

Comment 1-4: 4. Please consider doing a TUNEL assay on your sample to demonstrate apoptosis. This will support your claims.

Response 1-4: Thank you for the comment. Our previous study performed the TUNEL assay, which has been published in Nutrients[1]. These findings suggested that parental folate deficiency increased the apoptosis of neural cells both in the hippocampus and cerebral cortex of neonatal offspring, and the effects of maternal folate deficiency were more visible than paternal. Since these TUNEL results have already been published, we examined the expression of other apoptosis-related Bcl-2, Bax, and Caspase-3 indicators in offspring neonatal rat neural cells in this article. These observations in the TUNEL assay were consistent with the results of our present study.

References

  1. Ren, Q.; Zhang, G.; Dong, C.; Li, Z.; Zhou, D.; Huang, L.; Li, W.; Huang, G.; Yan, J. Parental Folate Deficiency Inhibits Proliferation and Increases Apoptosis of Neural Stem Cells in Rat Offspring: Aggravating Telomere Attrition as a Potential Mechanism. Nutrients 2023, 15, 2843.

Comment 1-5: Please consider adding an outcome where Hcy levels are investigated. It could be immunostaining, western blots, or HPLC. This will support your idea about Hcy.

Response 1-5: Thank you for the comment. We have previously tested for Hcy using a common and appropriate method (a competitive protein‐binding assay using Auto‐Chemistry Analyzer), and the results have been published[1]. In this article, in order to provide a clearer and more complete explanation of the regulatory roles of SAM and SAH, we cited our published Hcy results in the Results and Discussion sections. (page 3; line 105-108; page 8; line 259-261) The results showed that among the four groups, the D‐D group had the highest Hcy levels in the brain tissue of offspring rats, and the N‐N group had lower Hcy concentration than the D‐N group did. The findings indicated that the deficiency of parental folate increased Hcy concentrations, and the combined effects of folate deficiency in both parents played a more adverse part, followed by maternal folate deficiency and paternal folate deficiency.

  1. Ren, Q.; Zhang, G.; Dong, C.; Li, Z.; Zhou, D.; Huang, L.; Li, W.; Huang, G.; Yan, J. Parental Folate Deficiency Inhibits Proliferation and Increases Apoptosis of Neural Stem Cells in Rat Offspring: Aggravating Telomere Attrition as a Potential Mechanism. Nutrients 2023, 15, 2843.

Revised parts

Results (page 3; line 105-108)

“Our previous study demonstrated reduced folate levels in the neonatal rat offspring brain tissue, while parental folate deficiency increased Hcy concentrations. Furthermore, N-N group observed higher folate levels and lower Hcy concentrations in offspring brain tissue than the D-N group [24].”

Discussion (page 8; line 259-261)

“Research we have previously conducted in rats demonstrated that parental folate deficiency decreased folate levels and increased Hcy levels of offsprings’ brain tissue [24].”

Comment 1-6:   Please include a section about the limitations of the study in your discussion section.

Response 1-6: Thank you for the comment. We have added the limitations of the study in the Discussion section, and it has been indicated in red font.

Added parts

Discussion (page 9, line 303-311)

The study was a reproduction experiment in SD rats, with interventions in parental female and male rats fed folate-deficient and folate-normal diets, and related experiments were conducted to investigate the effects of folate-deficient intake by single-parent and both-parents on the offspring’s nerve cell apoptosis. Limitations of the research also exist. There must be multiple molecular mechanisms linking maternal and paternal folate levels to the offspring's nervous system, and more detailed and comprehensive studies should be designed to explore the contributing factors, thus providing the basis for folate intake by both parents in the periconceptional period, as well as for the prevention of neurological disorders in newborns.

Comment 1-7: Additionally, please add a paragraph on how you think folate deficiency in the males would lead to changes in the offspring. Do you believe it is due to epigenetic changes?

Response 1-7: Thank you for the comment. We have added the part suggested of the study in the discussion section, and it has been indicated in red font.

Added parts

Discussion (page 9, line 291-301)

Nevertheless, the relationship between paternal folate intake and foetal brain development has been very rarely reported. The findings of animal experiments demonstrated that paternal dietary folate deficiency may adversely affect birth outcomes in mice via epigenetic transmission involving sperm histone H3 methylation or DNA methylation, while dietary folate supplementation is critical for the offspring’s health [47]. Aatish Mahajan et al. demonstrated that an unbalanced folate in the paternal generation's diet could disrupt the progeny's methionine cycle [48]. Our findings also revealed that folate deficiency in either the paternal or maternal generation has consequences for the foetal nervous system and that SAM/SAH may be a regulatory factor. There are certainly other unexplored pathways, so more attention should be paid in the future to the association between both maternal and paternal folate intake and offspring health and the mechanisms involved.

References (page 15, line 541-546)

47. Lambrot, R.; Xu, C.; Saint-Phar, S.; Chountalos, G.; Cohen, T.; Paquet, M.; Suderman, M.; Hallett, M.; Kimmins, S. Low paternal dietary folate alters the mouse sperm epigenome and is associated with negative pregnancy outcomes. Nature communications 2013, 4, 2889, doi: 10.1038/ncomms3889.

48. Mahajan, A.; Sapehia, D.; Thakur, S.; Mohanraj, P.S.; Bagga, R.; Kaur, J. Effect of imbalance in folate and vitamin B12 in maternal/parental diet on global methylation and regulatory miRNAs. Scientific reports 2019, 9, 17602, doi: 10.1038/s41598-019-54070-9.

Reviewer 2 Report

In this paper, the authors explored the impact of maternal and paternal folate deficiency on neural cell apoptosis in neonatal offspring rats and the underlying pathogenic mechanism associated with SAM and SAH in brains.

The originality of this study is related to the exploration of the combined effect of maternal and paternal folate deficiency while most of previous investigations used “a single” model with maternal folate deficiency. Although the main finding reported in the study is that both maternal and paternal folate deficiency contribute to offspring pathogenesis the results indicated that maternal impact was stronger than paternal. This finding must be also reported in the Abstract section. What could be the explanation for this effect? Please, discuss it.

For statistical analysis one-way ANOVA was used. However, because of the low number of pups in a group (n = 5) suggesting high variability I guess a non-parametric test was required. For good statistical results at least 6 samples must be used. To increase clarity of the results, I would recommend using bar graphs that also include individual data points ie. bars by scatter plot with bar (for parametric method) or scatter plot (for non-parametric test).

It is important to indicate at weaning how many litters were collected and how the pups were distributed in groups.

It would be interesting to study the impact of parental deficiency during ontogenesis in different time windows as well as if there is some sex differences in offspring.

Minor: Indicate the significant differences with one symbol only (asterisk) vs a particular group with a line beneath the symbol placed above the two indicated groups with significant difference. 

 I am not qualified to assess the quality of English in this paper

Author Response

Point-by-point response to the reviewers’ comments:

Reviewer #2:

In this paper, the authors explored the impact of maternal and paternal folate deficiency on neural cell apoptosis in neonatal offspring rats and the underlying pathogenic mechanism associated with SAM and SAH in brains.

Response: Thank you for reviewing this article and providing valuable suggestions.

Comment 2-1: The originality of this study is related to the exploration of the combined effect of maternal and paternal folate deficiency while most of previous investigations used “a single” model with maternal folate deficiency. Although the main finding reported in the study is that both maternal and paternal folate deficiency contribute to offspring pathogenesis the results indicated that maternal impact was stronger than paternal. This finding must be also reported in the Abstract section. What could be the explanation for this effect? Please, discuss it.

Response 2-1: Thank you for your valuable suggestion. Our study focused on the combined and unilateral effects of maternal and paternal folate deficiency on neuronal apoptosis in neonatal rats whereas the results revealed that the effects were greatest in both parents, whereas the unilateral effect was manifested as greater in mothers than in fathers. The majority of study models have been single-parent designs, and the differences between the combined effects of both parental folate deficiencies and the impact of unilateral folate deficiency are inconclusive. Our observations in this study demonstrated that folate deficiency in both maternal and paternal affected neuronal apoptosis mediated by altered levels of SAM and SAH in rat foetal brains, with the greatest effect produced by both sexes acting together. The effect of single-sex in both the paternal and maternal progeny was stronger in the females than in the males. The reason for this may be the difference between paternal and maternal epigenetic inheritance, as maternal epigenetic reprogramming may occur at different times, more directly altering the offspring's environment (periconceptional environment) and affecting early embryonic reprogramming, whereas paternal epigenetic inheritance can only be transmitted via sperm. This part has been added to the Abstract and Discussion sections as suggested. It was indicated in red font.

Added parts

Abstract (Page 1, line 32-34)

“In conclusion, parental folate deficiency could induce the apoptosis of neural cells in neonatal offspring rats, while biparental folate deficiency had the greatest effect on offspring, and the unilateral effect was greater in mothers than in fathers.

Discussion (Page 9, line 277-281)

The reason for this may be the difference between paternal and maternal epigenetic inheritance, as maternal epigenetic reprogramming may occur at different times, more directly altering the offspring's environment (periconceptional environment) and affecting early embryonic reprogramming, whereas paternal epigenetic inheritance can only be transmitted via sperm [40].

Reference (page 14, line 520-523)

40. Sertorio, M.N.; César, H.; de Souza, E.A.; Mennitti, L.V.; Santamarina, A.B.; De Souza Mesquita, L.M.; Jucá, A.; Casagrande, B.P.; Estadella, D.; Aguiar, O., Jr.; et al. Parental High-Fat High-Sugar Diet Intake Programming Inflammatory and Oxidative Parameters of Reproductive Health in Male Offspring. Frontiers in cell and developmental biology 2022, 10, 867127, doi: 10.3389/fcell.2022.867127.

Comment 2-2: For statistical analysis one-way ANOVA was used. However, because of the low number of pups in a group (n = 5) suggesting high variability I guess a non-parametric test was required. For good statistical results at least 6 samples must be used. To increase clarity of the results, I would recommend using bar graphs that also include individual data points ie. bars by scatter plot with bar (for parametric method) or scatter plot (for non-parametric test).

Response 2-2: Thank you for the comment. As the study was a reproduction experiment in SD rats, the five samples from each group selected in the different experiments were from different litters and well represented. One-way analysis of variance (ANOVA) is the test designated for comparing the means of a study's target groups to identify if they are statistically different from the others. The principles of One-way ANOVA include randomization, normality, and variance chi-square[1]. Our animals were randomly grouped, and the data were subjected to normality and variance chi-square tests applicable to one-way ANOVA. We also reviewed the literature, and small data samples have used one-way ANOVA [2,3]. We have revised the statistical graphs in the Results and Discussion sections per your suggestion. The figures please see the attachment.

Reference

  1. Chatzi, A.; Doody, O. The one-way ANOVA test explained. Nurse researcher 2023, 31, 8-14, doi: 10.7748/nr.2023.e1885.
  2. Venediktova, N.; Solomadin, I.; Nikiforova, A.; Starinets, V.; Mironova, G. Functional State of Rat Heart Mitochondria in Experimental Hyperthyroidism. International journal of molecular sciences 2021, 22, doi: 10.3390/ijms222111744.
  3. Jo, W.; Min, B.S.; Yang, H.Y.; Park, N.H.; Kang, K.K.; Lee, S.; Chae, S.; Ma, E.S.; Son, W.C. Sappanone A Prevents Left Ventricular Dysfunction in a Rat Myocardial Ischemia Reperfusion Injury Model. International journal of molecular sciences 2020, 21, doi: 10.3390/ijms21186935.

Comment 2-3: It is important to indicate at weaning how many litters were collected and how the pups were distributed in groups.

Response 2-3: Thank you for the comment. The offspring rats were grouped the same as their mothers. At weaning, there were 24 offspring rats in each group: the D-D group from 9 litters, the D-N group from 10 litters, the N-D group from 6 litters, and the N-N group from 7 litters. The principle of distribution is that the five samples from the same group selected are from different litters, with a sex balance between males and females.

Comment 2-4: It would be interesting to study the impact of parental deficiency during ontogenesis in different time windows as well as if there is some sex differences in offspring.

Response 2-4: Thank you for the comment. Female and male rats were fed folate-normal and folate-deficient diets from 6 weeks of age, mated for a fortnight at sexual maturity in a 2:1 ratio, and received dietary treatment until females finished lactation and males finished mating. Regarding the study on the offspring, both female and male rats were involved. We reared the offspring rats to adulthood, during which time neurobehavioral testing was performed, including negative geotaxis test (PND5-PND9), cliff avoidance test (PND4-PND8), forelimb grip test (PND8, PND10, PND12, and PND14) and morris water maze test (PND45 and PND90). The results showed that parental folic acid deficiency inhibited the neurobehavioral development of offspring. These impacts on the neurodevelopment of offspring were most pronounced in the D-D group, followed by the D-N and the N-D groups. Our present study has not yet found a sex difference in the offspring.

Comment 2-5: Minor: Indicate the significant differences with one symbol only (asterisk) vs a particular group with a line beneath the symbol placed above the two indicated groups with significant difference

Response 2-5: Thank you for the comment. We have revised these figures as suggested. The figures please see the attachment.

Reviewer 3 Report

In article, authors investigated the apoptotic response of nerve cells induced by parental folate deficiency in neonatal rats. A number of factors (Bcl-2, Bax and caspase-3) were studied, the expression of which is of great interest for the characterization of the process. Overall, the study is well designed and the results are clearly presented. The authors presented a large amount of work, as well as comprehensive and reliable data.

During the review of the work, the following questions and comments arose:

Introduction: The manuscript is quite detailed, but the introduction is unclear as to how the model presented by the authors differs from the studies by Shao Y. et al. (2019). The authors need to emphasize the importance of the animal model and make it more concise. Authors are also encouraged to more clearly formulate the purpose and objectives of the work.

Methods: The methods section is well written and easy to understand. In the Results section, it would be nice if the authors presented Western immunoblotting data showing the dynamics of the expression of the studied factors.

 In the ultrastructural studies section, it is necessary to add the characteristics of intact control animals and the criteria by which pathological changes were assessed. On fig. 4b, there are no negative values of confidence intervals. What is it connected with?

Discussion section should include a broader view of the problem and is represented by a large amount of literature data. The authors need in conclusion to emphasize the importance of the level of folic acid in the body of the parents for the health and normal development of the nervous system in the offspring. It should also be pointed out that a deficiency of this vitamin can lead to serious disorders in the development and functioning of the nervous system in newborns, as well as indicate that further research is aimed at elucidating the molecular mechanisms underlying these changes and finding possible ways to prevent and treat such disorders.

English language needs some improvement

Author Response

Point-by-point response to the reviewers’ comments:

Reviewer #3:

In article, authors investigated the apoptotic response of nerve cells induced by parental folate deficiency in neonatal rats. A number of factors (Bcl-2, Bax and caspase-3) were studied, the expression of which is of great interest for the characterization of the process. Overall, the study is well designed and the results are clearly presented. The authors presented a large amount of work, as well as comprehensive and reliable data.

During the review of the work, the following questions and comments arose:

Response: Thank you for reviewing this article and providing valuable suggestions.

Comment 3-1: Introduction: The manuscript is quite detailed, but the introduction is unclear as to how the model presented by the authors differs from the studies by Shao Y. et al. (2019). The authors need to emphasize the importance of the animal model and make it more concise. Authors are also encouraged to more clearly formulate the purpose and objectives of the work.

Response 3-1: Thank you for the comment. There exists a difference in animal models in our study and the studies by Shao Y. et al. (2019): (1) Different animal species: we used Sprague-Dawley (SD) rats while C57BL/6 mice were used for that study. (2) Different experimental design: our study was a reproduction experiment in SD rats to investigate the effect of parental folate deficiency on neuronal apoptosis in the offspring, whereas that study by Shao Y. et al. only used C57BL/6 mice to produce neonatal mice to conduct the study, not the cross-generational study. Our study is a reproduction experiment in SD rats, commonly used experimental animals for their adaptability to the environment, rapid growth and development, high reproduction rate, and consistent genetics making them the preferred rodent for reproduction study[1]. Based on your suggestion, we have added these parts to the Materials and Methods sections, shown in red font.

References

  1. Hong, Y.; Gu, L.; Huang, Y.; Jin. F.; Chen, Q.; Tao, H.; Zhang, F.; Zhu, Y.; Xu, J.; Li, G.; Chen, W. Reproductive performance in SD rats and offspring growth and development. Occupation 160-2014, 30(22), 3215-3217, doi: 10.13329/j.cnki.zyyjk.2014.22.017.

Revised and added parts

Materials and Methods (page 9-10, line 314-317)

The study is a reproduction experiment in SD rats, commonly used experimental animals for their adaptability to the environment, rapid growth and development, high reproduction rate, and consistent genetics, making them the preferred rodent for reproduction studies [49].

References (Page 15, line 547-549)

49. Hong, Y.; Gu, L.; Huang, Y.; Jin. F.; Chen, Q.; Tao, H.; Zhang, F.; Zhu, Y.; Xu, J.; Li, G.; Chen, W. Reproductive performance in SD rats and offspring growth and development. Occupation 160-2014, 30(22), 3215-3217, doi: 10.13329/j.cnki.zyyjk.2014.22.017.

Comment 3-2: Methods: The methods section is well written and easy to understand. In the Results section, it would be nice if the authors presented Western immunoblotting data showing the dynamics of the expression of the studied factors.

Response 3-2: Thank you for the comment. Western immunoblotting can be performed on the whole brain tissue, while immunofluorescence assay can be performed the distribution of apoptosis-related genes in the hippocampus and central cortex regions of the brain. Moreover, immunofluorescence observations indicated that Bcl-2, Bax, and Caspase-3 expression differed between the hippocampus and the cortex. Therefore, the immunofluorescence assay was used in this study.

Comment 3-3: In the ultrastructural studies section, it is necessary to add the characteristics of intact control animals and the criteria by which pathological changes were assessed. On fig. 4b, there are no negative values of confidence intervals. What is it connected with?

Response 3-3: Thank you for the comment. The normal control group is shown in Figure 4a of the results in the D-D group. It appeared to have more normal mitochondrial structures and fewer swollen mitochondria exhibiting expanded matrix space, fragmented or disorganized cristae, and damaged membrane structure (M), fewer impaired myelinated axons with decompacted myelin (★), and more intact synaptic structures with a clear synaptic gap, a thickened synaptic membrane rich in dense material, and abundant synaptic vesicles in the presynaptic terminal (*). Based on your suggestion, we have revised these parts in the Results section, shown in red font. The confidence intervals you mentioned do not have a negative value because the minimum and the lower quartile are equal in the N-D and N-N groups, and the two lines coincide rather than the confidence interval having no minimum value.

Revised parts

Results (page 12, line 170-180)

As shown in Figure 4, N-N group appeared to have normal mitochondrial structures with a typical homogeneous staining pattern of the matrix, a clear visible double membrane and tight orderly cristae, and myelinated axons with a cozhangmpact layer of myelin lamellae and a greater number of synapses (P < 0.05). In contrast, the obvious pathological changes in neuronal cell bodies in D-D and D-N groups showed swollen mitochondria, exhibiting expanded matrix space, fragmented or disorganized cristae, damaged membrane structure, decompaction myelin exhibiting myelin sheaths surrounding axons loosening with slighted disrupted layers of myelin lamellae widespread and vacuoles as well as a fewer number of synapses (P < 0.05). Evidence from TEM experiments suggested that parental folate deficiency induced pathological changes in the ultrastructure of the hippocampus in neonatal offspring brains.

Comment 3-4: Discussion section should include a broader view of the problem and is represented by a large amount of literature data. The authors need in conclusion to emphasize the importance of the level of folic acid in the body of the parents for the health and normal development of the nervous system in the offspring. It should also be pointed out that a deficiency of this vitamin can lead to serious disorders in the development and functioning of the nervous system in newborns, as well as indicate that further research is aimed at elucidating the molecular mechanisms underlying these changes and finding possible ways to prevent and treat such disorders.

Response 3-4: Thank you for the comment. We've added these parts to the Discussion and Conclusion sections as your suggestion, shown in red font.

Discussion (page 9, line 282-301)

Since the 1980s, attention has been focused on the effects of maternal folate deficiency during pregnancy on fetal nervous system [41]. Adequate folic acid intake is particularly critical for pregnant women, whose folic acid requirements are 5-10 times higher than those of non-pregnant women. These increased requirements are essential to support the optimal growth and development of maternal and fetal tissues [42]. Maternal folate deficiency is considered to impact the neurodevelopment and brain health of the offspring, potentially leading to neurodevelopmental disorders, including neural tube defects [43]. However, the mechanisms by which maternal folate levels affect offspring neurodevelopment, including DNA methylation [44], oxidative stress [45], inflammation [46], etc., are complex and inconclusive. Nevertheless, the relationship between paternal folate intake and foetal brain development has been very rarely reported. The findings of animal experiments demonstrated that paternal dietary folate deficiency may adversely affect birth outcomes in mice via epigenetic transmission involving sperm histone H3 methylation or DNA methylation, while dietary folate supplementation is critical for the offspring’s health [47]. Aatish Mahajan et al. demonstrated that an unbalanced folate in the paternal generation's diet could disrupt the progeny's methionine cycle [48]. Our findings also revealed that folate deficiency in either the paternal or maternal generation has consequences for the foetal nervous system and that SAM/SAH may be a regulatory factor. There are certainly other unexplored pathways, so more attention should be paid in the future to the association between both maternal and paternal folate intake and offspring health and the mechanisms involved.

Conclusion (page 11, line 396-398)

This study highlights the importance of parental folate intake for the nervous system of the offspring, so preconception care should shift from mother-centered to parent-centered, and the mechanisms involved need to be further explored.

References (page 14-15, line 524-546)

41. De-Regil, L.M.; Fernández-Gaxiola, A.C.; Dowswell, T.; Peña-Rosas, J.P.           Effects and safety of periconceptional folate supplementation for                 preventing birth defects. The Cochrane database of systematic reviews         2010, Cd007950, doi: 10.1002/14651858.CD007950.pub2.

  1. Naninck, E.F.G.; Stijger, P.C.; Brouwer-Brolsma, E.M. The Importance of Maternal Folate Status for Brain Development and Function of Offspring. Advances in nutrition (Bethesda, Md.) 2019, 10, 502-519, doi: 10.1093/advances/nmy120.
  2. Caffrey, A.; McNulty, H.; Rollins, M.; Prasad, G.; Gaur, P.; Talcott, J.B.; Witton, C.; Cassidy, T.; Marshall, B.; Dornan, J.; et al. Effects of maternal folic acid supplementation during the second and third trimesters of pregnancy on neurocognitive development in the child: an 11-year follow-up from a randomised controlled trial. BMC medicine 2021, 19, 73, doi: 10.1186/s12916-021-01914-9.
  3. Bekdash, R.A. Early Life Nutrition and Mental Health: The Role of DNA Methylation. Nutrients 2021, 13, doi: 10.3390/nu13093111.
  4. Zhang, H.; Zhang, X.; Wang, Y.; Zhao, X.; Zhang, L.; Li, J.; Zhang, Y.; Wang, P.; Liang, H. Dietary Folic Acid Supplementation Attenuates Maternal High-Fat Diet-Induced Fetal Intrauterine Growth Retarded via Ameliorating Placental Inflammation and Oxidative Stress in Rats. Nutrients 2023, 15, doi: 10.3390/nu15143263.
  5. Priliani, L.; Oktavianthi, S.; Prado, E.L.; Malik, S.G.; Shankar, A.H. Maternal biomarker patterns for metabolism and inflammation in pregnancy are influenced by multiple micronutrient supplementation and associated with child biomarker patterns and nutritional status at 9-12 years of age. PloS one 2020, 15, e0216848, doi: 10.1371/journal.pone.0216848.
  6. Lambrot, R.; Xu, C.; Saint-Phar, S.; Chountalos, G.; Cohen, T.; Paquet, M.; Suderman, M.; Hallett, M.; Kimmins, S. Low paternal dietary folate alters the mouse sperm epigenome and is associated with negative pregnancy outcomes. Nature communications 2013, 4, 2889, doi: 10.1038/ncomms3889.
  7. Mahajan, A.; Sapehia, D.; Thakur, S.; Mohanraj, P.S.; Bagga, R.; Kaur, J. Effect of imbalance in folate and vitamin B12 in maternal/parental diet on global methylation and regulatory miRNAs. Scientific reports 2019, 9, 17602, doi: 10.1038/s41598-019-54070-9.

Round 2

Reviewer 2 Report

I find the manuscript improved and acceptable for publication in IJMS.